# DEEP LPPLS: FORECASTING OF TEMPORAL CRITICAL POINTS IN NATURAL, ENGINEERING AND FINANCIAL SYSTEMS VIA DEEP LEARNING

## ABSTRACT

The Log-Periodic Power Law Singularity (LPPLS) model offers a general framework for capturing dynamics and predicting transition points in diverse natural and social systems. In this work, we present two calibration techniques for the LPPLS model using deep learning. First, we introduce the Mono-LPPLS-NN (M-LNN) model; for any given empirical time series, a unique M-LNN model is trained and shown to outperform state-of-the-art techniques in estimating the nonlinear parameters $(t_c, m, \omega)$ of the LPPLS model as evidenced by the comprehensive distribution of parameter errors. Second, we extend the M-LNN model to a more general model architecture, the Poly-LPPLS-NN (P-LNN), which is able to quickly estimate the nonlinear parameters of the LPPLS model for any given time-series of a fixed length, including previously unseen time-series during training. The Poly class of models train on many synthetic LPPLS time-series augmented with various noise structures in a supervised manner. Given enough training examples, the P-LNN models also outperform state-of-the-art techniques for estimating the parameters of the LPPLS model as evidenced by the comprehensive distribution of parameter errors. Additionally, this class of models is shown to substantially reduce the time to obtain parameter estimates. Finally, we present applications to the diagnostic and prediction of two financial bubble peaks (followed by their crash) and of a famous rockslide. These contributions provide a bridge between deep learning and the study of the prediction of transition times in complex time series.

## 1 INTRODUCTION

The Log-Periodic Power Law Singularity (LPPLS) model provides a flexible and effective mathematical framework for representing the dynamics and for predicting temporal transition points which occur in a variety of natural and social systems (Sornette, 1998). Examples include the formation of financial bubbles followed by crashes in financial markets, the catastrophic material failure of systems subjected to constant or increasing stress loads (Anifrani et al., 1995), transient seismic and other geophysical activities preceding large earthquakes (Sornette & Sammis, 1995; Sornette et al., 2021; Mearns & Sornette, 2021), the formation of cusps in the surface curvature on the free surface of a conducting fluid in an electric field (Zubarev, 1998), vortex collapse of systems of point vortices (Leoncini et al., 2000), the finite-time formation of black holes according to the equations of General Relativity of the field metric coupled to a mass field (Choptuik, 1999a;b), the finite-time formation of fruiting bodies in models of aggregating micro-organisms (Rascle & Ziti, 1995), or in the more prosaic rotating coin (Euler's disk) (Moffatt, 2000).

LPPLS contains five words that combine three concepts, (i) singularity, (ii) power law and (iii) log-periodic, in order of importance.

The term "singularity" refers to the phenomenon where solutions to a large set of (ordinary or partial) differential equations exist only until a certain "critical" time, $t_c$. Consider the proportional growth equation $\frac{dp}{dt} = rp$. Here, $p$ can represent the population of a species, the size of a financial investment, or the GDP of a country, and $r$ is the growth rate. For a constant $r$, the solution is an exponentially growing function, $p(t) = p(0)e^{rt}$. To understand where a singularity can arise,

assume that the growth rate increases with $p$ as $r(p) = \eta p$, where $\eta > 0$ is a positive constant. In fact, this model reflects the growth of Earth's human population from around 1800 to the 1960s (Johansen & Sornette, 2001; Korotayev et al., 2006). This embodies a positive feedback of population on the growth rate. The solution to the new equation $\frac{dp}{dt} = \eta p^2$ is $p(t) = \frac{1}{t_c - t}$, where the critical time $t_c$ is determined from the initial condition $p(0) = \frac{1}{t_c}$. The solution exists for all times less than $t_c$, at which point a singularity occurs, i.e., $p(t)$ diverges as $t \to t_c$. In this case, the mathematical term for $t_c$ is a "movable singularity" (Bender & Orszag, 1999) because its value depends on the initial conditions (and on the structure of the equation). For $t > t_c$, there is no solution. The equation $\frac{dp}{dt} = \eta p^2$ is said to exhibit a finite-time singularity, a divergence in finite time. For more general equations, $p(t)$ can remain finite at $t_c$ and it is its first derivative that diverges, or higher order derivatives.

The presence of positive feedback (also known as pro-cyclicality in economics) in many dynamical systems leads to behaviors that transiently follow finite-time singular trajectories. A particularly attractive aspect of such mathematical descriptions is that the model contains the prediction of its own demise, the point where it approaches and passes $t_c$. This characteristic allows these models to naturally predict "regime changes" expected around $t_c$. Thus, the estimation of $t_c$ is crucial; having a framework to estimate it well could facilitate targeted interventions to either achieve desired outcomes or avoid undesirable ones.

The second concept "power law" is also illustrated by the above example in which $p(t)$ diverges according to a power law singularity function, here with exponent equal to $1$. Other finite time singularities can be logarithm like with $-\ln[t_c - t]$, essential like with $e^{\frac{1}{t_c - t}}$ or oscillatory with $\sin[\frac{1}{t_c - t}]$ and so on. One should not confuse this power law singularity with power law distributions.

The third concept "log-periodic" refers to the observable signature of the symmetry of discrete scale invariance (Sornette, 1998), which corresponds to a partial breaking of the symmetry of continuous scale invariance. Log-periodicity simply means that there is a periodicity in the logarithm of the distance to the critical point and, as a consequence a discrete hierarchy of times to $t_c$ which appears on top of the smooth power law structure. The periodicity in log-scale provides an anchor that can be used to improve prediction of $t_c$, similarly to frequency modulation in radio transmission where a tiny signal can be extracted by locking-in on a specific frequency. The simplest expression of a log-periodic function is $\cos[\omega \ln(t_c - t)]$ where $\omega$ does not have the meaning or dimension of an angular frequency. It rather represents the value of the ratio $\lambda$ of the discrete hierarchy of times to $t_c$ according to $\omega = \frac{2\pi}{\ln \lambda}$ (Sornette, 1998).

The simplest LPPLS model that combines the three concepts reads

$$\mathcal{O}(t) = A + B(t_c - t)^m + C(t_c - t)^m \cos(\omega \ln(t_c - t) - \phi) \tag{1}$$

where $\mathcal{O}(t)$ is the observable (for instance, the logarithm of the price for the diagnostic of a financial bubble). Among its 7 parameters, the three nonlinear parameters $t_c, m$ and $\omega$ play special roles. In the regime of interest here where $0 < m < 1$, we have $\mathcal{O}(t_c) = A$ and the singularity occurs in the first-order derivative of $\mathcal{O}(t)$. The sharpness of this singularity is controlled by the exponent $m$.

In standard calibration methods, one minimises the sum over all data points of the difference between the observable and the LPPLS model (1). Calculating the corresponding Hessian matrix and diagonalising it, one finds that parameters $t_c, \omega$ and $m$ in descending order are the most sloppy in a technical sense of the term (Brée et al., 2013; Filimonov et al., 2017): the log-likelihood function is the most flat along the direction $t_c$ and, unsurprisingly, the determination of the critical time is difficult (Demos & Sornette, 2017), if not unstable.

Given these limitations of standard calibration methods, our objective is to investigate the capabilities of neural networks (NNs) as universal function approximators, as highlighted by Hornik et al. (1989), in estimating the parameters of the LPPLS model. In this study, we present two NN architectures that are specifically designed for this purpose. The first architecture processes a single time-series to solve the inverse problem of optimally identifying the LPPLS parameters for that given time-series. Physics-Informed Neural Networks (PINNs), introduced by Raissi et al. (2019) and further developed by Faroughi et al. (2024), have proven effective in addressing inverse problems. They work by incorporating differential equations representing physical laws directly into the architecture of the NN, enabling them to learn solutions that inherently comply with the laws governing the systems they model. Our architecture, largely inspired by PINNs, incorporates the

LPPLS functional form in its learning process. Because this model operates on a single time-series, we term it the Mono-LPPLS-NN (M-LNN).

The second architecture we introduce is designed to operate in a conventional supervised manner, necessitating a large dataset of time-series (features) and their corresponding LPPLS parameters (labels). As no such empirical dataset exists, we generate a large corpus of synthetic time-series with known LPPLS parameters and introduce varying degrees of noise to obfuscate the LPPLS signal. We show that training on the synthetic dataset augmented by noise is sufficient to produce viable parameter estimates for unseen time-series and that its learning transfers to empirical datasets. Further, this class of models has the desirable property of any pre-trained NN model in that it can be deployed for near real-time inference. In our experimentation, we found that this class of models achieves estimations several orders of magnitude faster than existing state-of-the-art techniques on similar hardware configurations (see Table 2 in Appendix A.4 for details). Because this model is presented with many time-series, we term it the Poly-LPPLS-NN (P-LNN).

In the following sections, we describe the analyzed calibration techniques, which include the Levenberg-Marquardt (LM) method, which stands as the state-of-the-art for LPPLS parameter estimation, and our NN models, the M-LNN and P-LNN architectures. Our experimental design is then outlined, focusing on assessing the effectiveness of these models against the incumbent LM method by testing on a set of synthetic LPPLS series augmented by white noise and/or AR(1) noise. We highlight the performance of each model against both synthetic and real-world datasets. We conclude by discussing the broader implications of our results and proposing directions for future research.

## 2 METHODOLOGY

The state-of-the-art approach for estimating the $t_c$, $m$, and $\omega$ parameters of the LPPLS model is the LM method, which we detail in Appendix A.1. In this section, we focus on introducing two alternative NN calibration methods.

### 2.1 M-LNN MODEL

The M-LNN model is a feed-forward NN that estimates the nonlinear parameters of the LPPLS model from a specific time-series of arbitrary length. The M-LNN is a unique application where a new NN is trained for each empirical time series. This bespoke training approach has the potential to ensure that the model is adjusted to the characteristics of each unique dataset, perhaps offering more reliable parameter estimations. However, the trade-off here is that such an approach requires more computational resources.

#### 2.1.1 MODEL ARCHITECTURE

The architecture of the M-LNN is defined as follows (we also provide a comprehensive architectural diagram in Figure 1).

$$h_1 = \text{ReLU}(W_1 \cdot X + b_1), \quad h_2 = \text{ReLU}(W_2 \cdot h_1 + b_2), \quad Y = W_o \cdot h_2 + b_o, \quad (2)$$

where $X$ represents the input time series, $h_1$ and $h_2$ are the outputs from the first and second hidden layers, respectively, and $Y$ is the final output layer that yields the LPPLS parameter estimates. The weights $W_1, W_2$, and $W_o$ and biases $b_1, b_2$, and $b_o$ facilitate the transformation within the network layers. The ReLU (Rectified Linear Unit) function introduces non-linearity, enabling the network to capture and learn complex data patterns.

The Mean Squared Error (MSE) is used as the loss function $\mathcal{L}_{MSE}$, which measures the discrepancy between the actual time series data ($X$) and the LPPLS series derived from the estimated parameters (LPPLS($\hat{Y}$)). $\mathcal{L}_{MSE}$ is the same as the function $F(t_c, m, \omega, A, B, C_1, C_2)$ defined by expression (6) in Appendix A.1 with the change of notations $\mathcal{O}(\tau_i) \rightarrow X_i$ and $A - Bf_i - C_1g_i - C_2h_i \rightarrow$ LPPLS($\hat{Y}$)$_i$, so that LPPLS($\hat{Y}$)$_i$ corresponds to the element at the $i^{\text{th}}$ position from the LPPLS time-series derived from the estimated parameters.

To ensure the model's adherence to the empirical and theoretical constraints of the LPPLS framework (Saleur & Sornette, 1996), a penalty term is incorporated. This term bounds the pa-

rameter estimates within predefined ranges. The formulation of the penalty term should be informed by the appropriate application domain. For the application to the financial bubble diagnostics, these constraints have been discussed in e.g. Sornette & Johansen (2001); Sornette (2002); Sornette et al. (2015). The M-LNN penalty is expressed as $\mathcal{L}_{Penalty} = \alpha \sum_{k=1}^{3} (\max(0, \theta_{k,\min} - \theta_k) + \max(0, \theta_k - \theta_{k,\max}))$, where $\theta_{k=1} = t_c$, $\theta_{k=2} = m$ and $\theta_{k=3} = \omega$ and $\alpha$ is the penalty coefficient. $\theta_{k,\min}$ and $\theta_{k,\max}$ define the lower and upper bounds for the corresponding parameters. The specific parameter bounds we use are $t_2 - 0.2t_2 \le t_c \le t_2 + 0.2t_2$, $0.1 \le m \le 1$ and $6 \le \omega \le 13$. The time interval is normalised so that the beginning to the end of the time series is mapped to the unit segment $[0, 1]$. $t_2$ is the end time of the time series (here normalised to 1).

### 2.1.2 TRAINING PROCEDURE

Training of the M-LNN model (Eq. 2) shown in Figure 1 is conducted through a gradient descent optimization loop, where the objective is to minimize the combined loss ($\mathcal{L}_{MSE} + \mathcal{L}_{Penalty}$) over a specified number of epochs. It processes the input data to estimate the LPPLS parameters $t_c$, $m$, and $\omega$, which are subsequently used to calculate the linear parameters using Eq. 8 n Appendix A.1 and finally the estimated LPPLS time-series. The learning rate is set to $10^{-2}$. Prior to training, the dataset is min-max scaled to normalize the values and ensure consistent training dynamics. The training loop implements backpropagation to adjust the model parameters based on the total loss using the Adam optimizer. The model state is preserved at the epoch where the lowest total loss is achieved, indicative of the most accurate parameter predictions. This state is chosen as the best model fit.

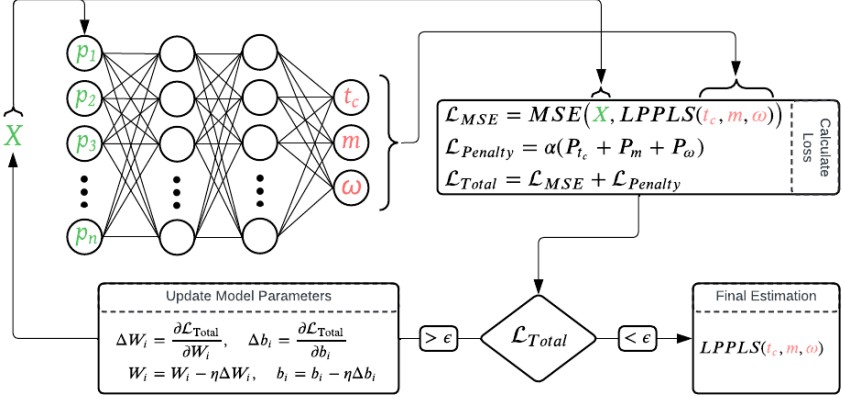

Figure 1: Diagram of the M-LNN architecture and training process. The network takes an input vector of length $n$, representing the time-series data for which the LPPLS parameters are to be estimated. The output layer, comprising three nodes, produces the estimated LPPLS parameters $t_c$, $m$, and $\omega$. These estimates are then utilized to construct the corresponding LPPLS time-series. The Mean Squared Error (MSE) between the input time series ($X$) and the LPPLS time-series is computed as the network's loss function ($\mathcal{L}$), with an additional penalty term to ensure the parameters remain within the designated bounds.

## 2.2 P-LNN MODELS

The P-LNN model adheres to a traditional machine learning paradigm in that it uses a structured set of training data with corresponding labels.

### 2.2.1 TRAINING SETS

Due to the absence of real-world datasets containing labeled LPPLS parameters, synthetic time-series data must be created to facilitate the training of the P-LNN models. Each variant within the P-LNN family of models undergoes training with a specific noise configuration and a set of randomly

generated LPPLS parameters (outlined in Table 1 in Appendix A.2). We train three distinct versions of the P-LNN model using differently augmented synthetic datasets. The first model, referred to as P-LNN-100K, is trained on synthetic data augmented with white noise. The second model, P-LNN-100K-AR1, utilizes synthetic data with AR(1) (autoregressive model of order 1) noise. Lastly, the P-LNN-100K-BOTH model is trained on a combination of synthetic data, augmented with both white and AR(1) noises, mixed in approximately equal proportions. This diversification in training setups allows us to explore the impact of noise characteristics on the model's performance. We describe the generation of these synthetic datasets in the following section.

### 2.2.2 GENERATION OF NOISY LPPLS TIME SERIES USED FOR TRAINING

Given a time-series of length $n$ generated by some known LPPLS parameters, $S = \{s_1, s_2, \ldots, s_n\}$, we generate a noisy version $S' = \{s'_1, s'_2, \ldots, s'_n\}$ with $s'_i = s_i + \eta_i$ where $\eta_i \sim \mathcal{N}(0, \alpha^2)$. The standard deviation $\alpha$ of the additive white noise quantifies its amplitude. Examples of obtained noisy synthetic LPPLS time series used for training data are shown in the supplemental material section in Figure 7.

Real time series described by the LPPLS model are often characterised by residuals that are not white noise (Zhou & Sornette, 2002a; 2003; Lin et al., 2014). The simplest extension of white noise to account for some memory is the AR(1) noise process. We thus also generate noisy synthetic LPPLS time series $s'_i = s_i + \eta_i$ with AR(1) noise defined by $\eta_t = \phi \cdot \eta_{t-1} + \varepsilon_t$, $t = 1, 2, \ldots, n$, where $\varepsilon_t$ follows a normal distribution $\mathcal{N}(0, \sigma^2)$. The variance $\sigma^2_\eta$ of the AR(1) noise $\eta_t$ is given by $\sigma^2_\eta = \frac{\sigma^2}{1-\phi^2}$.

### 2.2.3 MODEL ARCHITECTURE

The P-LNN model is configured with an input layer, four hidden layers, and an output layer. Forward propagation through the network is defined by the following:

$$h_1 = \text{ReLU}(W_1 \cdot X + b_1), \qquad h_2 = \text{ReLU}(W_2 \cdot h_1 + b_2),$$
$$h_3 = \text{ReLU}(W_3 \cdot h_2 + b_3), \qquad h_4 = \text{ReLU}(W_4 \cdot h_3 + b_4), \quad Y = W_5 \cdot h_4 + b_5, \tag{3}$$

where $W_i$ and $b_i$ are the weights and biases of the $i$-th layer, respectively. The ReLU function introduces non-linearity after each hidden layer, except for the output layer, which linearly combines the inputs from the last hidden layer to produce the output $Y$ which represents the 3 nonlinear parameters of the LPPLS model which we wish to estimate.

### 2.2.4 TRAINING PROCEDURE

The training process of the P-LNN model focuses on directly optimizing its ability to estimate LPPLS parameters, rather than comparing the original time-series with the estimated one derived from these parameters. This means that the loss function is chosen to be the sum of the squares of the three differences between the true and estimated LPPLS nonlinear parameters $t_c.m, \omega$. Initially, the NN is configured with specific architectural dimensions tailored to the input time-series data. Here, we use 252 nodes corresponding to time windows of $n = 252$ time points. Training occurs over 20 epochs, determined empirically to allow ample iterations for learning and weight updates to achieve satisfactory loss. Each epoch processes data in batches of 8 randomly selected noisy LPPLS time series. This value of 8 is chosen to balance memory resources and gradient descent efficiency. Batch processing introduces stochasticity into training, enhancing adaptability. The learning rate is set to $10^{-5}$. Before training, the dataset undergoes min-max scaling to normalize values and ensure consistent training dynamics. Loss curves for each model are included in Figures [8, 9, 10] in the appendix.

### 2.3 EXPERIMENTAL DESIGN

The objective of our experiment is to assess the capability of the parameter estimation techniques developed in this study. We focus on the LPPLS model parameters: critical time ($t_c$), exponent ($m$), and log-periodic oscillation frequency ($\omega$), under various noise conditions. We want to systematically explore a comprehensive range of these parameters to understand the robustness and accuracy

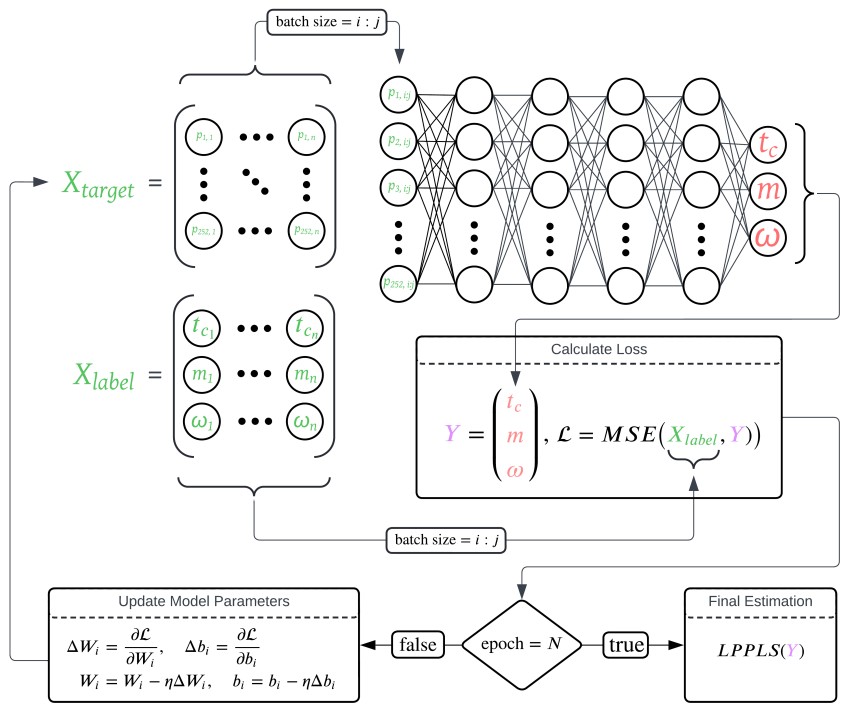

Figure 2: A diagram of the P-LNN architecture. The network accepts an input vector of length 252 representing a fixed-length time-series for which you want to obtain the LPPLS estimated parameters. Its output layer consists of three nodes which represent the LPPLS parameters $t_c$, $m$ and $\omega$. Next, we calculate the loss ($\mathcal{L}$) by taking the MSE between the network output and the true LPPLS parameters used to generate the batch of synthetic LPPLS series.

of the estimation techniques. To facilitate this, we randomly generate scenarios from the range of values for $t_c$, $m$, $\omega$, and noise levels, as summarized in Table 1 in Appendix A.2. We establish 250 unique scenarios providing a dense sampling of the parameter space for evaluation. For each unique scenario, we generate a time series data with known LPPLS parameters, overlaying them with noise as described in the methodology section 2.2.2. This setup allows for a precise ground truth against which the performance of each estimation technique is compared. By analyzing the estimation accuracy across all scenarios, we provide a comprehensive comparative analysis of the each model's effectiveness in parameter estimation across diverse conditions. We also record the wall-clock timing for each technique and report results in Table 2 in Appendix A.4.

## 3 RESULTS

### 3.1 SYNTHETIC DATA

The relative performances of the M-LNN and P-LNN models are assessed via the cumulative distribution function (CDF) of estimation errors. For each model, we generate four cumulative distribution functions (CDFs): one for the errors on $t_c$, another for the errors on $m$, a third for the error on $\omega$ and the fourth for the mean squared error (MSE) between the resulting calibrated LPPLS and the ground truth. Figure 3 shows these CDFs organized by rows according to the nature of the noise used to generate the synthetic noisy LPPLS time series used to train the models.

Using CDFs for performance comparison provides a full distributional view of all the characteristics of the competing models, which is better than usual statistics using average errors, quantiles and other point-wise metrics. The first important observation is that the M-LNN and P-LNN models tend to overperform the standard LM model, as seen from their stochastic dominance. For instance,

the M-LNN model is first-order stochastic dominant to the LM model for all three errors on the three nonlinear parameters.

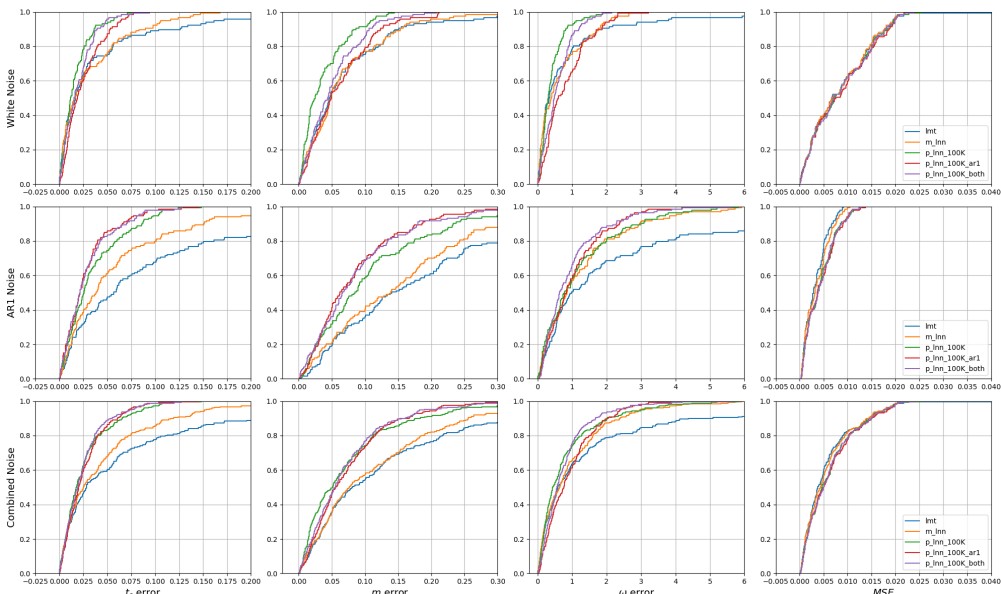

Figure 3: Cumulative distribution function (CDF) of absolute parameter error per estimation technique organized along rows for three classes of noise used in model training and ordered in four columns according to the variable whose error is quantified.

The P-LNN-100K model excels in minimizing smaller errors, demonstrating a sensitivity to parameter deviations. In contrast, the P-LNN-100K-BOTH model shows a greater tolerance for small to intermediate-sized errors but has fewer large errors, suggesting a trade-off in training with both white and AR(1) noise. This balance implies that, while training with just white noise prevents mislearning AR(1) structures, a mixed noise approach enhances the model's robustness against large errors.

The MSE distributions are less informative compared to the individual parameter distributions. The trained models do not distinguish themselves much in the resulting distance between calibrated and input time series. The MSE of the time series is a weak discriminating factor. That is, the models whose goal is to minimize the error between the observed time series and the time series constructed with the estimated LPPLS parameters do not necessarily yield accurate estimations of the actual parameters. This indicates that MSE minimization alone does not sufficiently differentiate model performance. This finding supports the notion of refining the calibration method by incorporating a penalty for errors in critical parameters like $t_c$.

## 3.2 EMPIRICAL DATA

To assess the transferability of the NN models to empirical data, we consider three datasets: (1) the daily adjusted closing price from the Nasdaq Composite Index from 1997 to 2000, a period encompassing the Dot-com bubble; (2) the daily adjusted closing price from the ProShares UltraSilver ETF from 2010 to 2011; (3) a dataset from the Veslemannen rockslide in 2019 (Kristensen et al., 2021). Each empirical dataset is resampled such that there are 252 observations preceding the critical time. This is to accommodate the current structure of the P-LNN model requiring an input size of 252. For each dataset, multiple calibrations are performed over a series of predefined observation windows within the time series data. Each window shifts the starting and ending value in order to systematically explore different temporal contexts. The ultimate goal is to identify the critical time $t_c$, which indicates an impending peak or crash in the time series, as predicted by each model.

We plot the actual time series data alongside the model fits and their extensions to visually assess the accuracy of each model's predictions. The visualization also includes probability density functions

(PDFs) of the predicted $t_c$ values across different calibration windows, providing a statistical view of each model's predictions. These plots allow us to compare the consistency and reliability of each modeling approach in identifying critical times in synthetic datasets designed to mimic real-world financial or physical systems exhibiting critical dynamics.

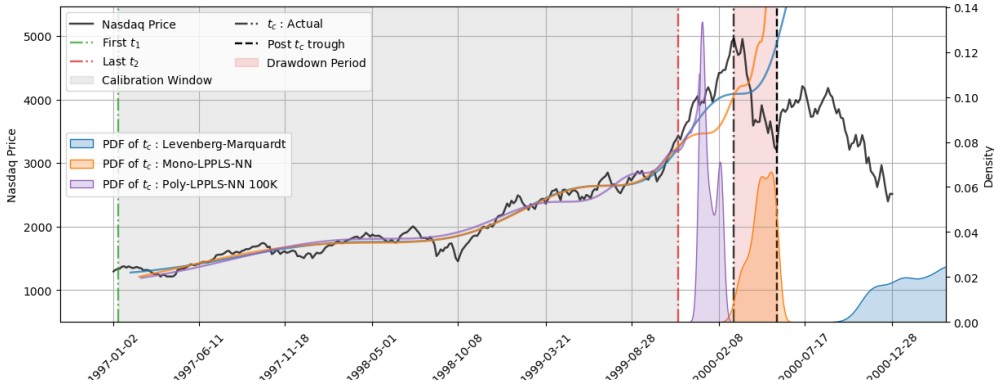

Figure 4: Dot-com bubble: the Nasdaq composite index price is plotted as a function of time (dark line) together with the LPPLS fits of the three competing models. The time interval of calibration goes from $t_1$ (green vertical dotted-dashed line) to $t_2$ (vertical red dotted-dashed line) and is represented in grey. We place ourselves in the "present" time $t_2$ so that all data to the right of $t_2$ is not seen and corresponds to the out-of-sample or future evolution that we aim to predict, and in particular, the actual $t_c$. Thus, all PDFs are constructed at time $t_2$. This realised critical time is interpreted to lie between the time of the price peak (black dotted-dashed vertical line) and the trough of its drawdown (black dashed vertical line). This range is indicated in shaded red colour. The M-LNN model (orange line) forecasts reasonably well the critical time, as evidenced from its PDF of $t_c$'s that overlaps largely the interval of the realised drawdown. The LM model (blue line) predicts $t_c$'s that are too late and with more variability in its predictions, reflected in its wider PDF spread. The P-LNN-100K model (purple line) captures the price trend well, with a narrower PDF indicating a higher concentration of $t_c$ predictions that are slightly early compared with the range of the actual critical time.

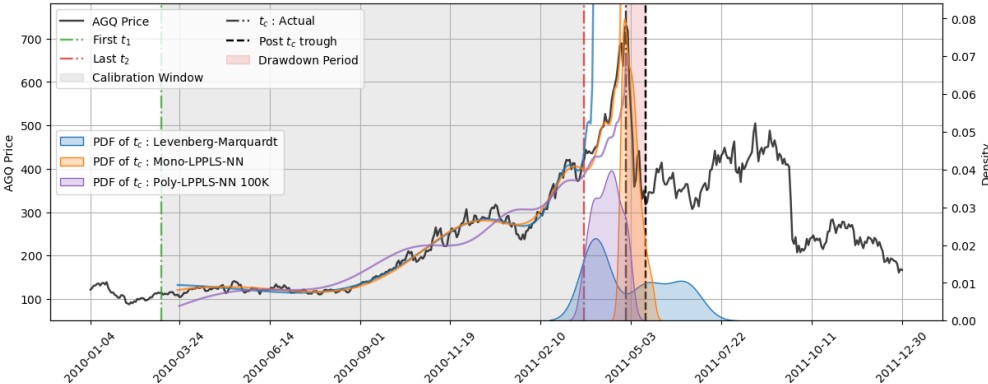

Figure 5: 2011 Silver bubble: the silver AGQ price is plotted as a function of time (dark line) together with the LPPLS fits of the three competing models. Same meaning of the different lines as in figure 4. In particular, the "present" time $t_2$ is indicated by the vertical red dotted-dashed line and the goal is to predict the subsequent behavior and especially the realized critical time $t_c$ of the crash. This time is well-defined by the vertical black dotted-dashed line indicating the time when the silver price peaked followed by an abrupt crash. The comparison of the three PDFs shows the significant superior performance of the NN models with the M-LNN model (orange line) providing an excellent prediction.

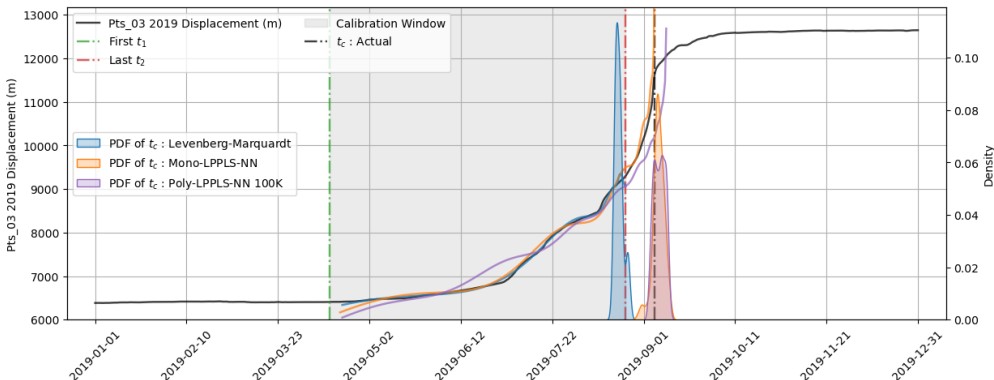

Figure 6: Vaslemannen rockslide Kristensen et al. (2021): the cumulative displacement of the rock-mass is plotted as a function of time. The rockslide occurs at the time indicated by the vertical black dotted-dashed line. Same meaning of the different lines as in Figure 4. The LM method predicts a $t_c$ much too early. In fact, it seems to interpret the latest acceleration just before present time $t_2$ as the imminent occurrence of the rockslide. In contrast, the two NN models give excellent forecasts with a slight superiority for the M-LNN method.

## 4 LIMITATIONS

A limitation of our study arises from the methodology used to generate both the training and testing datasets. We employed the same synthetic data generation technique for creating both sets of data. While this approach ensures consistency across training and testing phases, it also introduces potential biases towards the models' performance, as they are evaluated on data that may not adequately represent real-world variability and complexity. However, the good performance on the three empirical data sets, which are both unseen during the training and containing unknown noise structures, augurs well for the robustness of our NN models.

A second limitation is the relatively simple nature of the noise added to the LPPLS trajectories in the synthetic datasets used for the supervised training of P-LNN models. The residuals of LPPLS calibrations to real data are likely to be more structured than a white noise or even an AR(1) process as shown e.g. in Zhou & Sornette (2002b; 2003); Lin et al. (2014). Future works will need to extend the present methodology to noises that possess both long-range correlations and non-Gaussian fat tailed properties in the spirit of Ref. (Zhou & Sornette, 2002b). Indeed, one should always remember that the complete specification of a model includes both the model itself and the noise. Noise specification is often an afterthought of modellers while it should be an integral part of the model.

We have tested the performance of the trained NN models on three empirical datasets, with very encouraging results. However, this small sample size does not fully capture the diversity and complexity of real-world phenomena that could be modeled by NNs. A broader array of empirical datasets encompassing a more varied set of critical events may reveal novel properties of both the LPPLS structure and its residuals. This could inspire the development of more sophisticated LPPLS NN models, including generalised LPPLS formalisms in the spirit of Gluzman & Sornette (2002) and more powerful NN architectures.

## 5 CONCLUSION

Our research confirms the capability of NN models, specifically Mono-LPPLS-NN (M-LNN) and Poly-LPPLS-NN (P-LNN) models in estimating the parameters of the LPPLS model. This is demonstrated by our finding that they exhibit first-order stochastic dominance over the standard Levenberg-Marquardt (LM) method. Notably, the P-LNN-100K model trained with just white noise excelled in minimizing small errors, whereas the P-LNN-100K-BOTH model, trained with a blend of white noise and AR(1) noise, showed proficiency in reducing large errors, thus illustrating a balance in noise training methodologies.

The insights from our analysis underscore the need for a sophisticated approach in training these Poly NN models, particularly emphasizing the role of noise diversity in enhancing robustness and error. The observed variations in performance across different noise configurations indicate that tailored training strategies could significantly improve model adaptability and the accuracy of parameter estimation. One promising research path is the exploration of curriculum learning, as proposed by Bengio et al. (2009). This methodology posits that NNs may achieve superior generalization and accelerated convergence if the training examples are presented in an increasingly complex sequence. This strategy resonates with the idea of progressively intensifying noise complexity, which could potentially improve learning efficiency and model's ability to generalize.

Another research prospect stemming from this study involve the use of a larger variety of noise diversity and how that may impact parameter estimation error, especially by extending AR(1) noise with more sophisticated long-memory processes representing the rich dynamics of financial volatility. This could provide further insight into model behavior. Next, particularly for the Poly class of models, there exists a considerable scope for architectural innovation. Adopting advanced architectures like recurrent neural networks (RNNs) or transformers could mitigate the constraints associated with the fixed size of the input time-series data.

Finally, developing more intricate calibration techniques, potentially by embedding penalties for specific types of errors, could improve the precision in parameter estimation. Further empirical validation also emerges as an essential area of further research, necessitating the application of trained models to real-world datasets to corroborate their efficacy and utility in forecasting finite-time singularities across diverse domains.

Our research bridges the gap between NN methodologies and the analysis of temporal critical phenomena, offering promising directions for future investigations. The potential to predict finite-time singularities signalling regime shifts and transitions with improved accuracy has important consequences for both theoretical research and practical applications in natural, engineering and social sciences.

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

# A APPENDIX

## A.1 STANDARD CALIBRATION OF LPPLS VIA LEVENBERG-MARQUARDT (LM)

The standard calibration method proceeds as follows. First, Eq. 1 is modified so as to reduce the complexity of the calibration process by reformulating the linear parameter $C$ and non-linear parameter $\phi$ in terms of two linear parameters $C_1 = C \cos \phi$ and $C_2 = C \sin \phi$, yielding

$$\mathcal{O}(t) = A + B(f) + C_1(g) + C_2(h). \tag{4}$$

where

$$\begin{aligned} f &= (t_c - t)^m \\ g &= (t_c - t)^m cos\big(\omega \ln (t_c - t)\big) \\ h &= (t_c - t)^m sin\big(\omega \ln (t_c - t)\big) \end{aligned} \tag{5}$$

The 3 nonlinear parameters $\{t_c, m, \omega\}$ and 4 linear parameters $\{A, B, C_1, C_2\}$ are calibrated by minimising

$$F(t_c, m, \omega, A, B, C_1, C_2) = \frac{1}{n} \sum_{i=1}^{n} \Big[ \mathcal{O}(\tau_i) - A - Bf_i - C_1 g_i - C_2 h_i \Big]^2 \tag{6}$$

where $n$ denotes the length of the input time-series and $f_i = f(\tau_i), g_i = g(\tau_i)$ and $h_i = h(\tau_i)$. Following (Filimonov & Sornette, 2013), we proceed in two steps. First, at fixed values of $t_c, m, \omega$, the 4 linear parameters are estimated by solving the optimization problem:

$$\{\hat{A}, \hat{B}, \hat{C}_1, \hat{C}_2\} = \arg \min_{A,B,C_1,C_2} F(t_c, m, \omega, A, B, C_1, C_2) \tag{7}$$

which is done analytically by solving the following matrix equation obtained from the first-order condition of (7)

$$\begin{pmatrix} N & \sum f_i & \sum g_i & \sum h_i \\ \sum f_i & \sum f_i^2 & \sum f_i g_i & \sum f_i h_i \\ \sum g_i & \sum f_i g_i & \sum g_i^2 & \sum g_i h_i \\ \sum h_i & \sum f_i h_i & \sum g_i h_i & \sum h_i^2 \end{pmatrix} \begin{pmatrix} \hat{A} \\ \hat{B} \\ \hat{C}_1 \\ \hat{C}_2 \end{pmatrix} = \begin{pmatrix} \sum \ln p_i \\ \sum f_i \ln p_i \\ \sum g_i \ln p_i \\ \sum h_i \ln p_i \end{pmatrix} \tag{8}$$

Reporting the values $\hat{A}(t_c, m, \omega), \hat{B}(t_c, m, \omega), \hat{C}_1(t_c, m, \omega), \hat{C}_2(t_c, m, \omega)$ in expression (6) gives the reduced loss function for the 3 nonlinear parameters

$$F_1(t_c, m, \omega) = \min_{A,B,C_1,C_2} F(t_c, m, \omega, A, B, C_1, C_2) = F(t_c, m, \omega, \hat{A}, \hat{B}, \hat{C}_1, \hat{C}_2) \tag{9}$$

Next, the 3 nonlinear parameters can be determined by solving the following nonlinear optimization problem:

$$\{\hat{t_c}, \hat{m}, \hat{\omega}\} = \arg \min_{t_c, m, \omega} F_1(t_c, m, \omega) \tag{10}$$

The LM algorithm is then used to find the best estimation of the three nonlinear parameters $t_c, m, \omega$ as it offers an efficient interplay between gradient descent and Gauss-Newton methods to accelerate convergence. (Moré, 1978) discusses the algorithm's efficiency in navigating the parameter space of nonlinear models in order to calibrate their parameters.

## A.2 PARAMETERS FOR GENERATING NOISY LPPLS SERIES

Table 1: Parameter ranges and values for generating synthetic LPPLS time series for P-LNN Models. Each synthetic dataset is split into 100,000 training samples and 33,333 validation samples. Parameter values for $t_c$, $m$, and $\omega$ are randomly chosen within the specified ranges. The noise amplitude represents a percentage of the total function range, which is always 1, as the training data is rescaled to be in the range $[0, 1]$.

| Model | Noise Type | Noise Amplitude |
|---|---|---|
| P-LNN-100K | White | 0.01 to 0.15 |
| P-LNN-100K-AR1 | AR1 | 0.01 to 0.05 |
| P-LNN-100K-BOTH | White and AR(1) | White: 0.01 to 0.15, AR1: 0.01 to 0.05 |

| Parameters for Generating Synthetic Data: | |
|---|---|
| Parameter | Range or Value |
| $t_c$ | $t_2$ to $t_2 + 50$ (days) |
| $m$ | 0.1 to 0.9 |
| $\omega$ | 6 to 13 |
| Autoregressive Coefficient ($\phi$) | 0.9 |

## A.3 SYNTHETIC TRAINING DATA PLOTS

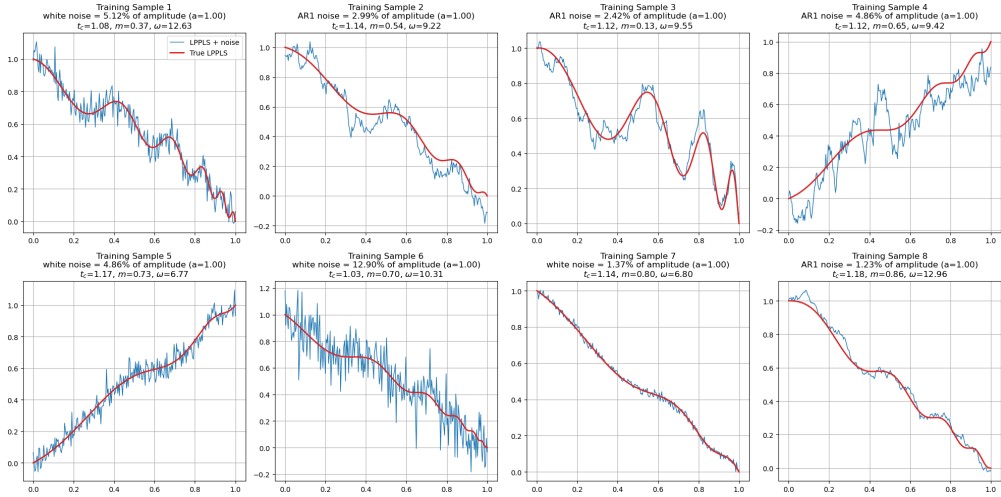

Figure 7: Random samples of LPPLS training data with white & AR(1) noise augmentation.

## A.4 WALL-CLOCK TIMINGS

Table 2: Average Execution Times and Standard Deviations for Each Calibration Technique. Each method performed 250 calibration trials conducted on identical hardware configurations to ensure comparability. Wall-clock times were recorded, averaged, and their standard deviations calculated to assess the variability of each technique. Notably, the P-LNN-* models demonstrate computational efficiency that is orders of magnitude faster than the LM method

| Method | Average Time (s) | Standard Deviation (s) |
|---|---|---|
| LM | 3.5795 | 8.4000 |
| M-LNN | 11.5002 | 4.7944 |
| P-LNN-100K | 0.0052 | 0.00052 |
| P-LNN-100K-AR1 | 0.0052 | 0.0021 |
| P-LNN-100K-BOTH | 0.0050 | 0.00061 |

## A.5 COMPUTE

For model training, we used Google Cloud Platform's Tesla V100-SXM2 GPUs with 16GB of memory. Each variation of P-LNN model required approximately 1.5 hours of compute time on these GPUs. Additionally, we conducted hyperparameter tuning to optimize model performance based on our loss metric. This involved a grid search across various combinations of epochs, learning rates, and batch sizes, structured as 125 configurations. Due to the use of a reduced dataset size for these experiments, the training time for each configuration ranged from 15 to 30 minutes. The combined compute time for the main model training was approximately 4.5 hours, and the hyperparameter tuning phase cumulatively required approximately 46 hours.

## A.6 P-LNN-* TRAINING

This set of figures illustrates the training and validation loss curves for the Poly models (white, AR(1), and combined noise) over 20 epochs. Each curve demonstrates a consistent decrease in loss, indicating effective learning and generalization capabilities of the models.

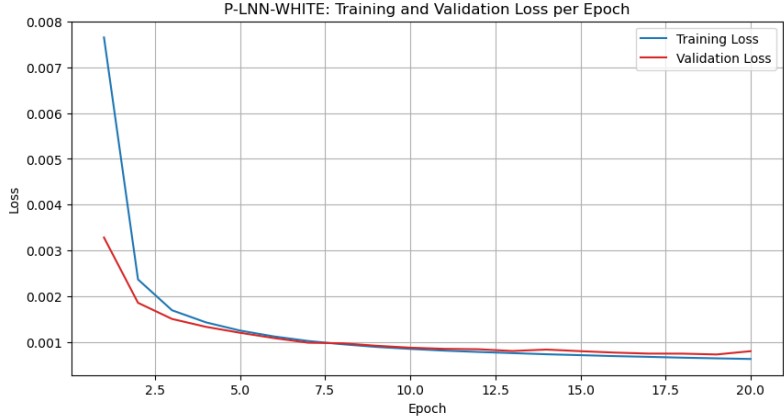

Figure 8: The loss curve for the P-LNN-100K model trained with white noise.

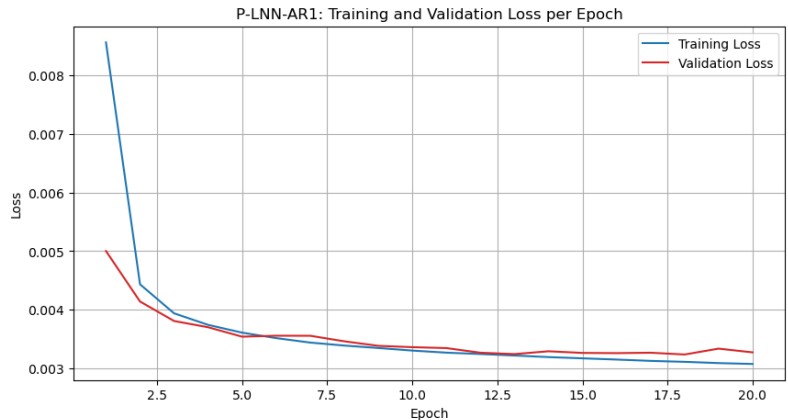

Figure 9: The loss curve for the P-LNN-100K model trained with AR(1) noise.

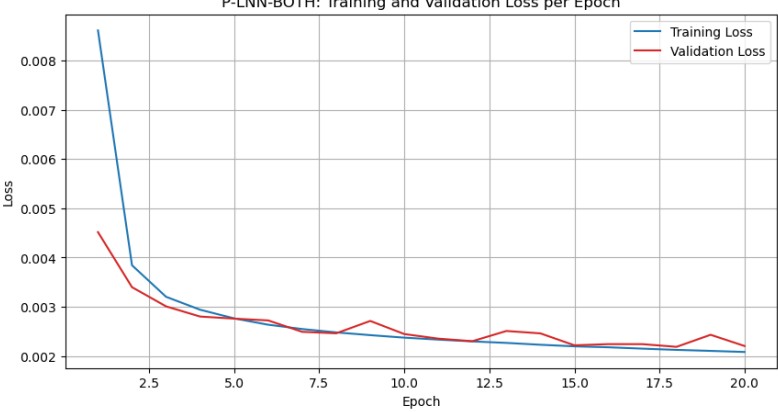

Figure 10: The loss curve for the P-LNN-100K model trained with both white and AR(1) noise.

