# OpenReview forum: "Deep LPPLS: Forecasting of temporal critical points in natural, engineering and financial systems via deep learning"
_ICLR.cc/2025/Conference — ICLR 2025 Conference Withdrawn Submission_

### Official Review · Reviewer_esVF · 2024-10-15

**Soundness:** 2
**Presentation:** 3
**Contribution:** 2
**Rating:** 3
**Confidence:** 4

**Summary:**

This paper introduces two deep learning models, Mono-LPPLS-NN (M-LNN) and Poly-LPPLS-NN (P-LNN), that estimate the critical parameters of the Log-Periodic Power Law Singularity (LPPLS) model. This is a mathematical model that is used to predict transition points in complex systems, such as financial bubbles and ensuing crashes. The M-LNN model is trained on individual time-series, while the P-LNN model is trained across multiple datasets. The authors benchmark their approaches against Levenberg–Marquardt, a classical iterative optimization algorithm for fitting nonlinear models to data. The paper argues that both M-LNN and P-LNN models reduce parameter estimation errors and provide faster predictions compared to traditional methods. The models were tested on synthetic data and real-world events like the 2000 Nasdaq dot-com bubble and a rockslide, both cases in which the exact timing of the critical event parameter must be precisely identified.

**Strengths:**

I have not seen the LPPLS problem come up before in a scientific machine learning context, and it is a timely and worthwhile problem to study. The paper is clearly written, and walks the reader very clearly through exactly what the authors did and how/why they did it. A range of testing datasets was used, which span different domains and application areas relevant to practitioners.

**Weaknesses:**

**Scope:** The LPPLS system is an interesting model, but it does not appear to be widely-applied. While the authors describe the individual terms and parameters in this model in detail, I’m not seeing what makes this a particularly important problem, or what properties of the model make it difficult to fit using conventional methods.

**Methodology:** The authors are fitting three parameters to time series data. This is essentially a low-dimensional curve fitting exercise, and it is not obvious to me why we need to use neural networks. I would expect that a variety of methods, like gradient descent or sequential parameter fitting, would perform equally well on this problem. The authors claim motivation from PINNs, but a signature feature of PINNs, the use of autodiff gradients to fit derivative terms in PDE, isn’t used here because the LPPLS system is not an ODE/PDE

I do not feel that the experiments are rigorous or extensive enough to justify the performance claims. The synthetic data seems very simple, it’s just an LPPLS problem with autocorrelated white noise. The size of this dataset does not need to be highlighted so prominently in the paper, and it’s not obvious that the model improves from more data, given the low-dimensional fitting problem. There is only one benchmark model, and no ablations.

A major concern is the P-LNN training procedure doesn’t directly compare the fitted LPPLS dynamics to the true dynamics, but rather directly compares the three fitted parameters to the exact ground truth values used to make the training data. This is a simple regression problem that I would expect linear regression to readily solve. It’s not obvious to me why this was done, but this undermines the utility of this method for real time series, where the parameters of the generating process are unknown in advance. I would expect that, in the real data experiments, the model is essentially mapping the input time series onto the whichever training example is most similar to it.

I have some more mild concerns about other aspects of the experiment design. There is no clear train/test splitting (because the authors are treating this as a direct fitting problem). Particularly with time series, including forecasting of crashes, it is important to avoid causal leakage with time series cross-validation. Related to this concern, I find the testing on real data a bit artificial. A lot of choices are made regarding aligning the time series shapes, time to critical events, and the range of parameter values searched, and so I don’t necessarily find these results robust.

**Reproducibility:** For a this problem and application, there is no reason why the code shouldn’t be available.

**Questions:**

+ The penalty term (line 166) seems strange to me. How are the upper bounds selected? It seems like prior knowledge of the test data set might be leaking into the training here.

+ Why is so much compute needed? Given that the widest MLP layer is 252 units, I don't see where there would be substantial computational cost.

---

### Official Review · Reviewer_E4Zo · 2024-10-26

**Soundness:** 2
**Presentation:** 3
**Contribution:** 2
**Rating:** 3
**Confidence:** 3

**Summary:**

This paper introduces two innovative machine learning approaches to calibrate the Log-Periodic Power Law Singularity (LPPLS) model, a framework for capturing dynamics and predicting transition points in complex time series, such as those in natural and social systems. This model leverages a large dataset of synthetic LPPLS time series with various noise structures, resulting in faster parameter estimation and more robust performance compared to existing methods. The paper also includes practical applications, analyzing two financial bubbles and a notable rockslide, demonstrating the model’s utility for diagnosing and predicting critical points in complex systems.

**Strengths:**

This paper employs neural network tools to estimate the parameters of the LPPLS model, demonstrating superior accuracy and robustness compared to the standard Levenberg-Marquardt (LM) method. The authors clearly delineate their methodology and experimental design. The proposed technique plays a pivotal role in forecasting critical points, thereby holding significant relevance across natural, engineering, and economic systems.

**Weaknesses:**

This method primarily integrates multi-layer neural networks with the LPPLS model without incorporating significantly novel designs. Consequently, its innovation appears somewhat deficient. Furthermore, the experimental validation section fails to adequately substantiate the advantages of the proposed method, and it lacks a comprehensive explanation as to why employing neural network strategies outperforms conventional methodologies.

**Questions:**

1. Eqution 1 mentioned in the main text involves a small number of parameters; thus, there should be various methods to estimate these unknown parameters. Why adopting a connectionist approach such as neural networks would yield better results than classical methods?

2. Using only the Levenberg-Marquardt (LM) as the baseline method seems to be insufficient.

3. For sequential data, methods like RNNs seem to possess a more powerful capability in extracting dynamic information. Why is a regular feedforward neural network chosen here?

4. The text lacks an explanation of the setting of hyperparameters and a sensitivity analysis.

---

### Official Review · Reviewer_VFFi · 2024-10-31

**Soundness:** 1
**Presentation:** 2
**Contribution:** 2
**Rating:** 3
**Confidence:** 4

**Summary:**

The paper introduces two neural network models, M-LNN and P-LNN, for estimating LPPLS parameters to predict critical transition points in complex systems directly from time series data. M-LNN is a fully connected network trained on individual time series, while P-LNN is a fixed-length model pre-trained on synthetic data to generalize across time series. Both models aim to improve parameter estimation over traditional methods for estimating LPPLS parameters like Levenberg-Marquardt. The models are tested with two different noise models and on three empirical datasets to assess their effectiveness in recovering ground truth parameters and in predicting critical time points.

**Strengths:**

- The introduction is well-written, and the motivation behind LPPLS is clearly laid out.
- The overall idea to estimate LPPLS parameters with ML techniques also seems promising.
- The real-world applications and comparison of different noise models are interesting.

**Weaknesses:**

- Penalty term: the motivation for this is a bit too short. Literature is provided for the specific financial domain (but without further comments), but since this is designed as a general-purpose deep learning architecture, what is a more general motivation for finding these priors/penalties if literature on this specific topic does not exist?

- Architecture: this is perhaps my biggest concern: the M-LNN model is just a fully connected feedforward NN, which takes in the entire input time series, and doesn’t contain any specific inductive biases for handling sequential or time-series data. The conclusion describes RNNs as "advanced architectures" for addressing fixed input-size constraints in time series data. However, RNNs (including LSTMs and GRUs) have been established as foundational models in time series analysis for decades, and would (to my mind) be the obvious choice here (and get rid of the fixed input size as a bonus). Other specific sequence models like TCNs or even Transformers would be more justifiably framed as advanced models and could be tested here as well.

- This problem becomes even more pronounced when moving toward the P-LNN model: the network only accepts an input vector of length 252. This fixed length undermines the flexibility expected from a pre-trained architecture and requires resampling of empirical datasets.
- Underscoring this concern, in the experiments, it says “Each empirical dataset is resampled such that there are 252 observations preceding the critical time. This is to accommodate the current structure of the P-LNN model requiring an input size of 252.” There are no details on how this sampling is accommodated. How much data is “thrown away”? Do you require some specific imputation methods to fit the sequence length precisely? As far as I could see, no details are provided in the appendix.

- Figure 2 outlines a standard supervised training setup using basic gradient descent, which feels too simplistic for a major ML conference. This could be moved to the supplement, especially given the redundancy with Figure 1 (which is also quite simplistic). Additionally, the optimizer details are missing, and the figures suggest only vanilla gradient descent may have been used, which would be suboptimal for training neural networks.

- Figure 3: text and legend+linewidth could be slightly larger to enhance readability.

- The results section would benefit from a more thorough discussion of the findings. Currently, the results are mostly described, without an analysis of what the results imply for the model’s performance and limitations. For instance, in the discussion, you say “We have tested the performance of the trained NN models on three empirical datasets, with very encouraging results”, while neither the fits look very convincing, nor were the predictions of t_c so strong for the NASDAQ data.

- You also mention multiple times that your NN models outperform the traditional LM method, but a more in-depth comparison (perhaps with statistical significance tests of the results in Fig.3) would make this claim more explicit.

Given the current limitations, I believe the paper may not yet be ready for a leading ML conference. While the paper addresses an interesting problem, it needs a stronger emphasis on architectural choices, methodological depth, and more thorough evaluation to meet the standards of a high-impact ML venue. Otherwise, the paper might be better suited for a different venue that focuses less on ML techniques and more on applications.

**Questions:**

If not too pressed for space, some motivation in the paragraph about “power laws” could be added, since this kind of universal scaling with power-law behaviors of observables near critical points/phase transitions is widely studied in physics, or in the context of the “critical brain hypothesis” in neuroscience. Likewise, in the paragraph about “log-periodic” behavior, the motivation is a bit abstract, and could also benefit from a real-world example (e.g. synchronization of trading behavior before financial bubbles)

Typo: p.9: augurs instead of argues

---

### Official Review · Reviewer_1iMj · 2024-11-02

**Soundness:** 2
**Presentation:** 3
**Contribution:** 2
**Rating:** 3
**Confidence:** 3

**Summary:**

The paper provides a presentation of the Log-Periodic Power Law Singularity (LPPLS) model enhanced by neural networks. While it gives detailed explanations of two proposed models, the Mono-LPPLS-NN (M-LNN) and Poly-LPPLS-NN (P-LNN), the technical descriptions could benefit from simplification for broader comprehension. Additionally, critical steps in training procedures and model parameters lack transparency and are challenging to reproduce without additional details. It seems that the title overclaim the ability of the rather simple deep learning  NN model.

**Strengths:**

The paper provides an in-depth explanation of the model architectures and training procedures for both M-LNN and P-LNN, as well as the noise structures used to augment synthetic data.

**Weaknesses:**

While the paper suggests broad applicability in predicting critical points in natural and engineering systems, such as rockslides and material failures, only three datasets are used for empirical validation, with two focusing on financial bubbles (the Nasdaq Dot-com bubble and the Silver ETF bubble). The third example, a rockslide, is not sufficiently detailed to confirm that the model can reliably generalize beyond financial contexts. Without substantial empirical data from natural or engineering domains, claims about the model’s predictive capability in these areas seem overstated.

**Questions:**

1. Is there any detailed data set for engineering or natural problems that this method can predict temporal critical points? like phase transition or the earthquake Seismic waves? if not, it is overselling of the present model ability

2. The paper heavily relies on synthetic data with simplified noise configurations (e.g., white noise, AR(1) noise) that might not adequately represent real-world complexities. It would be great to use some open data sets other than just the economic data

---

### Note · Authors · 2024-11-27

**Comment:**

Thank you for the valuable feedback. We feel that it is best to withdrawal, address comments and find a more suitable venue for applied techniques as a reviewer mentions.

Best,
1661 Authors

**Withdrawal Confirmation:**

I have read and agree with the venue's withdrawal policy on behalf of myself and my co-authors.